HOXC6 promotes migration, invasion and proliferation of esophageal squamous cell carcinoma cells via modulating expression of genes involved in malignant phenotypes

Tang Li
Cao Yong
Song Xueqin
Wang Xiaoyan
Li Yan
Yu Minglan
Li Mingying
Liu Xu
Huang Fang
Chen Feng
Wan Haisu whssyzx@swmu.edu.cn
Experimental Medicine Center, The Affiliated Hospital of Southwest Medical University , Luzhou , China
Tao Shi-Cong
Electronic publication date: 2019 Mar 14
Publication date: 2019
Volume: 7
Electronic Location ID: e6607
Received 2018 Nov 2; Accepted 2019 Feb 10
Copyright: © 2019 Tang et al.
Copyright year: 2019
Copyright holder: Tang et al.
License: This is an open access article distributed under the terms of the Creative Commons Attribution License, which permits unrestricted use, distribution, reproduction and adaptation in any medium and for any purpose provided that it is properly attributed. For attribution, the original author(s), title, publication source (PeerJ) and either DOI or URL of the article must be cited.
License URL: https://creativecommons.org/licenses/by/4.0/

Keywords: ESCC, Homeobox, HOXC6, Migration, Invasion, Proliferation

Funding: Joint Program on the Science and Technolog Collaboration of Southwest Medical University and the Government of Luzhou City 2018LZXNYD-PT04 Science Technology Support Plan Projects of Luzhou 2015LZCYD-S02 Project Program of Southwest Medical University 2017-ZRQN-005 This study was supported by the grants from the Joint Program on the Science and Technolog Collaboration of Southwest Medical University and the Government of Luzhou City (No. 2018LZXNYD-PT04), the Science Technology Support Plan Projects of Luzhou (No. 2015LZCYD-S02) and the Project Program of Southwest Medical University (2017-ZRQN-005). The funders had no role in study design, data collection and analysis, decision to publish, or preparation of the manuscript.

==============================
Background

HOXC6 is a member of the HOX gene family. The elevated expression of this gene occurs in prostate and breast cancers. However, the role of HOXC6 in esophageal squamous cell carcinoma (ESCC) remains largely uninvestigated.

Methods

The expression of HOXC6 was examined by immunohistochemistry, quantitative real-time PCR and immunoblotting assays. The lentivirus-mediated expression of HOXC6 was verified at mRNA and protein levels. Wound healing and Matrigel assays were performed to assess the effect of HOXC6 on the migration and invasion of cancer cells. The growth curving, CCK8, and colony formation assays were utilized to access the proliferation capacities. RNA-seq was performed to evaluate the downstream targets of HOXC6. Bioinformatic tool was used to analyze the gene expression.

Results

HOXC6 was highly expressed in ESCC tissues. HOXC6 overexpression promoted the migration, invasion, and proliferation of both Eca109 and TE10 cells. There were 2,155 up-regulated and 759 down-regulated genes in Eca109-HOXC6 cells and 95 up-regulated and 47 down-regulated genes in TE10-HOXC6 cells compared with the results of control. Interestingly, there were only 20 common genes, including 17 up-regulated and three down-regulated genes with similar changes upon HOXC6 transfection in both cell lines. HOXC6 activated several crucial genes implicated in the malignant phenotype of cancer cells.

Discussion

HOXC6 is highly expressed in ESCC and promotes malignant phenotype of ESCC cells. HOXC6 can be used as a new therapeutic target of ESCC.

Introduction

Esophageal squamous cell carcinoma (ESCC) incidence is the eighth highest, and mortality is the sixth highest, of all cancers worldwide (Cai et al., 2015). Despite the fact that many efforts have been made to improve the diagnosis and therapy of ESCC, the overall 5-year survival rate remains disappointing, and it is still one of the most fatal malignancies (Kashyap et al., 2009; Zhang, 2013). The main reason for this is that ESCC is usually in the advanced stages at diagnosis (Kashyap et al., 2009; Zhang, 2013). To address this problem, it is necessary to identify potential molecular markers that may be used for the diagnosis and therapy of ESCC.

Homeobox-containing gene family comprises approximately 200 transcription factors which share a 183 base pairs long DNA region called homeobox in their coding sequences, and the homeobox encodes a 61 amino acids homeodomain (HD) with characteristic fold (Cantile et al., 2011). The HOX genes are a subgroup of homeobox-containing genes encoding transcription factors that confer segmental identities in the process of development. In humans, there are 39 HOX genes clustered into four different groups (HOXA, HOXB, HOXC, and HOXD). HOX genes are crucial to the regulation and control of the processes important to development, such as receptor signaling, apoptosis, motility, differentiation, and angiogenesis (Zhang et al., 2013). Many HOX genes have been found to be either activated or repressed in the process of cancer development. Aberrant expression of HOX genes has also been reported in a variety of cancers, such as colorectal (Kanai et al., 2010; Liao et al., 2011), breast (Hur et al., 2014; Shaoqiang et al., 2013), prostate (Chen et al., 2012b), glioblastoma (Costa et al., 2010), and lung (Abe et al., 2006) cancers.

HOXC6 is a member of the HOX family, and its aberrant expression has been verified in a variety of cancers, such as prostate (Ramachandran et al., 2005), breast (Hussain et al., 2015), nasopharyngeal carcinoma (Chang et al., 2017), gastric (Zhang et al., 2013), and ovarian (Tait et al., 2015) cancers. HOXC6 overexpression promoted cell migration, invasion and proliferation, where decreased HOXC6 expression reversed the facilitation effect on gastric cancer cells (Chen et al., 2016). In hepatocellular carcinoma, HOXC6 overexpression promoted cell proliferation, while siRNA-mediated HOXC6 down-regulation not only inhibited proliferation and migration but also increased 5-FU chemosensitivity (Sui et al., 2016). Ji et al. (2016) also found that silencing of HOXC6 expression inhibited the proliferation of colorectal cancer cells. Collectively, these studies suggest that HOXC6 might be involved in tumor initiation and progression. In the case of ESCC, it is predicated that HOXC6 may be highly expressed in ESCC tissues compared to adjacent normal counterparts (Du et al., 2014). However, the role of HOXC6 in ESCC has not been fully investigated. Here, we report that HOXC6 functions as an oncogene in ESCC cells via up-regulation of genes associated with the malignant phenotype. HOXC6 is a candidate molecular marker for both the diagnosis and treatment of ESCC.

Materials and Methods

Cell lines and cell culture

The ESCC cell lines, Eca109 and TE10, were purchased from the Shanghai Institute of Cell Biology, Chinese Academy of Sciences (Shanghai, China). 293FT cell line was obtained from Shanghai Tongpai biotechnology co. LTD (Shanghai, China). Eca109 and TE10 cells were cultured in RPMI 1640 medium (Gibco, Grand Island, NY, USA) and 293FT cells was maintained in DMEM (Gibco, Grand Island, NY, USA). All mediums were supplemented with 10% fetal bovine serum (Gibco, Grand Island, NY, USA), 100 μ/ml of penicillin and 100 μ/ml of streptomycin. All cells were cultured in a 37 °C, 5% CO2 incubator.

Patients and specimens

Esophageal squamous cell carcinoma tissues and adjacent normal counterpart specimens were obtained from patients with ESCC who were treated with surgery between January 2017 and August 2017 at the Department of Thoracic Surgery, the Affiliated Hospital of Southwest Medical University (Luzhou, China). A portion of each specimen was immediately frozen in liquid nitrogen for qRT-PCR and western blotting assays. Another portion was immediately fixed in neutral formalin buffer and embedded into paraffin for histopathological observation. The present study was approved by the Ethics Committee of the Affiliated Hospital of Southwest Medical University (NO. K2018002-R). Written informed consents for this study were obtained from all patients.

RNA extraction and qRT-PCR

Total RNA from cells and tissues was extracted with Trizol (Invitrogen, Carlsbad, CA, USA; Thermo Fisher Scientific Inc., Waltham, MA, USA) following the manufacturer’s instructions. For each specimen, 500 ng of total RNA was used for reverse-transcription using the PrimeScript™RT reagent Kit with gDNA Eraser (TaKaRa Bio Inc., Tokyo, Japan). The reaction conditions of reverse-transcription were 15 min at 37 °C, and 5 s at 85 °C. qRT-PCR examination was performed using SYBR Premix Ex Taq II (Tli RNaseH Plus) (TaKaRa Bio Inc., Tokyo, Japan). The primer sequences are shown in Table 1. The reaction conditions were as follows: 95 °C for 30 s, followed by 40 cycles of 95 °C for 5 s, then 60 °C for 34 s. Reactions were carried out in an Applied Biosystems 7500 Real Time PCR System (Applied Biosystems, Foster City, CA, USA; Thermo Fisher Scientific Inc., Waltham, MA, USA). The expression of GAPDH was used as an internal control and the RNA expression level of each gene was evaluated using the 2−ΔΔct method. All specimens were examined in triplicate.

Table 1 The primer sequences for qRT-PCR.

Gene	Primer sequence (5′–3′)	
Forward	Reverse	
HOXC6	ACAGACCTCAATCGCTCAGGA	AGGGGTAAATCTGGATACTGGC	
MMP14	CGAGGTGCCCTATGCCTAC	CTCGGCAGAGTCAAAGTGG	
SPARC	AGCACCCCATTGACGGGTA	GGTCACAGGTCTCGAAAAAGC	
FN1	CGGTGGCTGTCAGTCAAAG	AAACCTCGGCTTCCTCCATAA	
INHBA	CAACAGGACCAGGACCAAAGT	GAGAGCAACAGTTCACTCCTC	
SEMA3C	TAACCAAGAGGAATGCGGTCA	TGCTCCTGTTATTGTCAGTCAGT	
HEY1	ATCTGCTAAGCTAGAAAAAGCCG	GTGCGCGTCAAAGTAACCT	
SGK1	GCAGAAGAAGTGTTCTATGCAGT	CCGCTCCGACATAATATGCTT	
SMAD9	CTGTGCTCGTGCCAAGACA	TGGAAAGAGTCAGGATAGGTGG	
WNT6	GGTGCGAGAGTGCCAGTTC	CGTCTCCCGAATGTCCTGTT	
BRCA2	CACCCACCCTTAGTTCTACTGT	CCAATGTGGTCTTTGCAGCTAT	
SAMD9	GCAACCATCCATAGACCTGAC	AATAGTGCCATTGGTACGTGAAT	
AKAP9	CACGGCATAAGGGAGAAATGG	GCTGTCTCTGTAGAGCACACT	
SATB1	CCAGGTTGGAAAGTGGAATCC	GGGGCAACTGTGTAACTGAAT	
CFH	GTGAAGTGTTTACCAGTGACAGC	AACCGTACTGCTTGTCCAAAA	
JAK2	TCTGGGGAGTATGTTGCAGAA	AGACATGGTTGGGTGGATACC	
MCTP1	AGTTTACGCCTATCAGACCTACA	GATCGCTCAACCCGTTGGAAT	
ANXA10	GCTGGCCTCATGTACCCAC	CAAGCAGTAGGCTTCTCGC	
DAPP1	CAGCCTTTGATTGGAAGCGAG	TGTGAACCCGGACAGATTCAT	
GAPDH	CTCTGACTTCAACAGCGACACC	CTGTTGCTGTAGCCAAATTCGTT	

Immunohistochemistry

Formalin-fixed paraffin-embedded blocks were prepared into tissue sections. The sections were treated with 3% H2O2 for 10 min after routine deparaffinization in xylene and rehydration in decreasing concentrations of ethanol (100, 95, 85, and 75%). Then, sections were heated in citrate sodium for approximately 3 min for antigen retrieval. To block nonspecific reactions, the sections were incubated with 10% normal goat serum for 20 min after antigen retrieval. Then, the sections were incubated with a mouse monoclonal antibody against human HOXC6 (Santa Cruz, dilution, 1:200) overnight at 4 °C. After primary antibody incubation, the streptavidin/peroxidase amplification kit (ZSGB-Bio, Beijing, China) was used for the HOXC6 antigen-antibody reaction. Then, sections were treated with diaminobenzidine to visualize the appearance of HOXC6 signal. To quantitate the expression of HOXC6, two specialists in pathology independently scored the immunohistochemical signals according to intensity (0–3) and extent (0–100%). The staining intensity was categorized as follows: 0, negative (−); 1, weak (+); 2, moderate (++); and 3, strong (+++). The expression of HOXC6 was calculated as the product of the intensity and extent scores (IHC score).

Western blotting

Lysates of cells and tissues were prepared using pre-cooled RIPA buffer (Cell Signaling, Danvers, MA, USA) supplemented with proteinase inhibitor (Roche, Indianapolis, IN, USA). The total protein concentration was tested using the BCA protein assay kit (Thermo Fisher Scientific Inc., Waltham, MA, USA). Total protein lysate (20 μg) of each sample was separated using sodium dodecyl sulfate-polyacrylamide gel electrophoresis and then transferred to polyvinylidene difluoride membranes (Millipore, Billerica, MA, USA). Next, the membranes were soaked in 5% skim milk power to block nonspecific reactions and then incubated with mouse monoclonal antibody against human HOXC6 (Santacruz, dilution, 1:1,000) overnight at 4 °C. Membranes were then incubated with the corresponding secondary antibody for 30 min. Immunoreactive protein was visualized using the Western Lighting Chemiluminescence Reagent Plus (PerkinElmer, Waltham, CA, USA). β-actin (Santacruz, dilution, 1:1,000) was used as internal control.

Establishment of cell lines stably expressing the exogenous HOXC6

The lentiviral HOXC6-expression vector pCDH-HOXC6 was constructed by cloning the human HOXC6 gene into pCDH-NEO vector which is generated by replacing the copGFP gene with genemycin resistance gene of pCDH-CMV-MCS-EF1-copGFP (JiRan, Shanghai, China). The primer sequences including the XhoI and EcoRV restriction enzyme sites are shown in Table 2. The lentiviral particles were produced by transfecting either pCDH-HOXC6 or pCDH-NEO with psPAX2 and pMD2.G into 293FT cells. The culture supernatant was harvested and subsequently infected Eca109 and TE10 cells. The stable cells were selected by genemycin at a concentration of 500 μg/ml (Thermo Fisher Scientific Inc., Waltham, MA, USA).

Table 2 The primer sequences for PCR amplification.

Gene fragment	Primer sequence (5′-3′)	
PCDH-CMV-MCS-EF1	Forward	GCGTGGATATCTAAGTCGACAATCAACCTCTGG	
	Reverse	AATATCTCGAGGGTGGCGTCTAGCGTAGGCG	
Genemycin resistance gene	Forward	GTCTACTCGAGGCCACCATGATTGAACAAGATGGATTGCAC	
	Reverse	GCTATGATATCTCAGAAGAACTCGTCAAGAAGGC	
HOXC6	Forward	CCCGGTCTAGAGCCACCATGAATTCCTACTTCACTAACCCTTCC	
	Reverse	TTGGCGGATCCTCACTCTTTCTGCTTCTCCTCTTC	

Wound healing and matrigel invasion assay

Cells were seeded in 96-well plates and allowed to achieve 90–100% confluence. The Essen Bioscience 96-pin wound maker (Essen BioScience, Ann Arbor, MI, USA) was used to create a uniform scratch in each well. For the wound healing assay, cells were incubated in RPMI 1640 medium supplemented with 2% fetal bovine serum. For the invasion assay, cells were covered with 50 μl of Matrigel solution (2.4 mg/ml Matrigel in normal growth medium, BD Biosciences, San Jose, CA, USA) that was allowed to gel at 37 °C for 1 h. Then, an additional 100 μl/well of normal growth medium was used to overlay the matrix. The wound of each well was monitored and images were taken at 2 h intervals in an IncuCyte live-cell analysis system (Essen BioScience, Ann Arbor, MI, USA). The wound width was calculated using the software provided.

Growth, CCK8, and colony formation assays

Cells were plated in 96-well plates (3,000 cells/well). Cell growth was monitored and images were taken at 2 h intervals in an IncuCyte live-cell analysis system. Growth curves were calculated from confluence measurements using image analysis software. For the Cell Counting Kit-8 assay (Dojindo Molecular Technologies, Inc., Kumamoto, Japan), optical density was measured to determine cell activity at a wavelength of 450 nm on a microplate reader at 0, 24, 48, and 72 h. For the colony formation assay, stably infected cells were seeded in six-well plates (200 cells/well) and the growth medium was changed at 3-day intervals. After 10 days of incubation at 37 °C with 5% CO2, cells were fixed with 4% paraformaldehyde and stained with freshly prepared, diluted Giemsa stain for 20 min. Colony number was counted after excess dye was washed off with double-distilled water.

RNA-seq analysis

Total RNA was isolated with Trizol and the polyadenylated mRNAs were enriched using Dynabeads Oligo (dT) 25 beads. Three replicates were created for RNA-seq library construction using the NEBNext Ultra Directional RNA Library Prep Kit for Illumina (NEB, Ipswich, MA, USA), following the manufacturer’s instructions. Illumina HiSeq Xten was used to perform 150 bp pair-end sequencing. We mapped all RNA-seq data to the GRCh37.p13 genome from GENCODE (Harrow et al., 2012) by HISAT2 (version 2.1.0) (Kim, Langmead & Salzberg, 2015) with default parameters. To identify differentially expressed genes (DEGs), we aggregated the read counts at the gene level using HTseq (Anders, Pyl & Huber, 2015), and then identified DEGs using R package DESeq2 (Love, Huber & Anders, 2014). Genes were considered significantly differentially expressed when the |log2 (fold-change)| > 1 and adjusted p < 0.05. DEGs were subjected to enriched GO categorization using the R package clusterProfiler (Yu et al., 2012) with a q-value <0.05.

Bioinformatic analysis

Gene Expression Omnibus (GEO) is a public functional genomics data repository supporting MIAME-compliant data submissions. We utilized this tool to investigate gene expression. GEO is available at https://www.ncbi.nlm.nih.gov/geo/.

Statistical analysis

All experimental data are presented as the mean ± standard deviation from two to three separate experiments. Wound healing Matrigel invasion and growth curve assays were conducted in at least 12 repeats for every separate experiment. CCK8 and colony formation assays were conducted in triplicate. Data was analyzed using SPSS19.0 software. Comparisons between groups were performed using Student’s t-test, and differences were considered statistically significant at p < 0.05.

Results

HOXC6 was highly expressed in ESCC tissues

HOXC6 is highly expressed in many cancer types (Figs. 1A–1I), including ESCC (Figs. 1G–1I). This suggests that HOXC6 may be a critical factor in cancer development. However, the role of this gene in ESCC is not clear. We investigated the expression of HOXC6 in ESCC cells. We first investigated the expression of HOXC6 using clinical tumor samples. A total of 32 paired samples from patients that underwent ESCC resection were used to verify the expression of HOXC6 by IHC and qRT-PCR. Representative images of the HOXC6 IHC assays are shown in Fig. 2A. The IHC score of HOXC6 in tumor tissues was significantly higher than in adjacent normal counterparts (Fig. 2B). The relative HOXC6 mRNA expression level in tumor tissues was also significantly higher than in adjacent normal counterparts (Fig. 2C). We randomly selected three paired samples to analyze the expression level of the HOXC6 protein by western blotting, and the results showed that HOXC6 in tumor tissues was highly expressed when compared with adjacent normal counterparts (Figs. 2D and 2E). These results suggest that HOXC6 expression may play a role in the malignancy of ESCC.

Figure 1 HOXC6 is highly expressed in many cancer types.

Microarray data retrieved from gene expression omnibus (GEO) repository were utilized to analyze HOXC6 expression at mRNA level in various cancers including (A) lung cancer, (B) colorectal cancer, (C) prostate cancer, (D) soft-tissue sarcoma, (E) renal carcinoma, (F) gastric cancer and (G–I) ESCC.

Figure 2 HOXC6 was highly expressed in ESCC tumor tissues.

(A) Representative images and (B) IHC score of HOXC6 from immunohistochemical analysis. (C) Relative HOXC6 mRNA expression levels in tumor tissues compared with normal tissues as measured by qRT-PCR. Each of the columns in the bar graph represents the value of one tumor sample as compared with that of its normal control. The expression of GAPDH was used as the internal control. HOXC6 mRNA expression level was evaluated using the 2−ΔΔct method. All specimens (n = 32) were examined in triplicate. (D) Representative images of western blot, β-actin has been used as an internal control. (E) The quantification of the relative expression level of HOXC6 as determined by normalized gray values. **p < 0.01; ***p < 0.001.

HOXC6 promoted ESCC cell migration and invasion

To explore how HOXC6 affects the malignant phenotype of ESCC cells, we introduced the HOXC6 gene into Eca109 and TE-10 cells via lentiviral-mediated transfection, generating cell lines stably expressing HOXC6 (Eca109-HOXC6 and TE10-HOXC6) as well as cell lines transfected with empty vector (Eca109-NEO and TE10-NEO). Fig. 3A shows the lentiviral vector pCDH-HOXC6. The expression level of HOXC6 was confirmed by both qRT-PCR (Fig. 3B), western blotting (Figs. 3C and 3D) and immunocytochemistry assay (Fig. 3E). Since the migration and invasion of tumor cells is critical for tumor angiogenesis and metastasis, we assessed the effect of HOXC6 on ESCC cell migration and invasion using wound healing and Matrigel invasion assays. By calculating wound healing width and invasion width of the scratch wound, we found that HOXC6 overexpression significantly increased cell migration speed in Eca109-HOXC6 and TE10-HOXC6 cells compared to controls (Figs. 4A–4F). The Matrigel invasion assay demonstrated that HOXC6 significantly improved the invasive capacity of the ESCC cells (Figs. 5A–5F). These results suggest that HOXC6 may act as a facilitator in promoting ESCC cell migration and invasion.

Figure 3 Lentiviral vector and verification of HOXC6 overexpression.

(A) The recombinant lentiviral vector pCDH-HOXC6. (B) The quantification of relative HOXC6 mRNA expression levels as measured by qRT-PCR. GAPDH has been used as the internal control. (C) The result of western blot and (D) the quantification of relative HOXC6 protein expression levels as determined by the normalized gray values. β-actin has been used as an internal control. (E) HOXC6 expression in HOXC6-transfected cells and their controls examined by immunocytochemical staining. **p < 0.01; ***p < 0.001.

Figure 4 HOXC6 promoted ESCC cell migration.

Wound healing assay was performed to evaluate the migration of stably infected cells. Representative images from the wound healing assay in (A) Eca109 and (B) TE10 stable cells. (C–D) The wound healing width of Eca109-HOXC6 was significantly wider than in the Eca109-NEO cells. Similarly, (E–F) the effect of HOXC6 expression on the ability of wound healing of the two ESCC cell lines. There were at least 12 replicates included for each cell line. These data demonstrated that HOXC6 promoted ESCC cell migration. ***p < 0.001.

Figure 5 HOXC6 promoted ESCC cell invasion.

Matrigel invasion assay was used to assess cell invasion capacity. Representative images from the Matrigel invasion assay in (A) Eca109 and (B) TE10 stable cells. By calculating the invasion width, the invasion capacity of (C–D) Eca109-HOXC6 and (E–F) TE10-HOXC6 cells was greater than the controls. These data demonstrate that HOXC6 effectively promoted ESCC cell invasion. *p < 0.05; **p < 0.01.

HOXC6 promoted ESCC cell proliferation

To assess the effect of HOXC6 on ESCC cell proliferation, we utilized a real-time monitoring assay to measure the growth rates of Eca109-HOXC6 and TE10-HOXC6 as well as their controls. Growth curves were constructed from data points acquired at 2 h intervals. Results showed that the growth of Eca109-HOXC6 and TE10-HOXC6 was significantly faster than that of controls (Figs. 6A and 6B). Furthermore, CCK8 and colony formation assays were also performed. The OD450 values of Eca109-HOXC6 and TE10-HOXC6 were higher than that of controls when measured after 72 h (Figs. 6C and 6D). In addition, the results of the colony formation assay showed that Eca109-HOXC6 and TE10-HOXC6 generated more colonies than the controls (Figs. 6E–6H). Collectively, these results suggest that HOXC6 may increase the proliferation and colony formation of ESCC cells.

Figure 6 HOXC6 promoted ESCC cell proliferation.

Growth curves were constructed via real-time monitoring. (A) The growth curve of Eca109-HOXC6 and Eca109-NEO cell lines. (B) The growth curve of TE10-HOXC6 and TE10-NEO cell lines. (C–D) The growth curves as determined by OD450 values using CCK8 kit and (E–H) colony formation assay was also used to evaluate the colony formation ability of these cell lines. CCK8 and colony formation assays were conducted in triplicate. Overall, these data suggest that HOXC6 promoted ESCC cell proliferation. *p < 0.05; **p < 0.01; ***p < 0.001.

RNA-seq analysis identified genes regulated by HOXC6

As a member of the homeobox gene family, HOXC6 possesses the characteristics of a transcription factor that can bind to a specific sequence in the genome and regulate the expression of related genes. To identify the downstream targets and the regulatory network of HOXC6, RNA-seq was conducted to compare changes of mRNA expression patterns following HOXC6 transfection. The raw RNA-seq data in this article has been uploaded to GEO repository and the accession number is GSE121976. The expression levels of individual genes were measured by sequence counts. Genes with at least a two-fold change in expression were viewed as either up-regulated or down-regulated genes. The global changes in mRNA expression patterns are shown in Fig. 7A and 7B for Eca109-HOXC6 and TE10-HOXC6. As shown in Fig. 7C and 7D, there were 2,155 up-regulated and 759 down-regulated genes in Eca109-HOXC6 cells compared with Eca109-NEO cells. In addition, there were 95 up-regulated and 47 down-regulated genes in TE10-HOXC6 cells compared with TE10-neo cells. Interestingly, there were only 20 common genes, including 17 up-regulated and three down-regulated genes with similar changes upon HOXC6 transfection in both cell lines indicating the cell-context dependent function of HOXC6 in different cell lines. The common regulated genes are shown in Fig. 7E in the format of a table. There are 14 common up-regulated genes, many of these genes have been reported to be implicated in the development of various types of cancers. There are also three common down-regulated genes, however, the function of these genes has not been well characterized. We further confirmed the results of RNA-seq by examining the expression of HOXC6-modulated genes using qRT-PCR. As shown in Figs. 7F and 7G, changes in expression level of these genes were consistent with the RNA-seq results.

Figure 7 RNA-seq analysis of the interactive genes of HOXC6 in Eca109 and TE10 stable cells.

(A–B) The volcano diagrams show the global changes in mRNA expression patterns for Eca109 and TE10 stable cells. (C–D) The Venn diagrams show the common up-regulated genes and down-regulated genes and (E) the common regulated genes are also shown in the table. (F–G) The relative expression of genes regulated by HOXC6 at the mRNA level were identified by qRT-PCR. (H–I) GO analysis showed various functional groups of genes regulated by HOXC6 and (J–K) KEGG pathway analysis showed functional pathways for Eca109 and TE10 stable cells.

HOXC6 up-regulated critical genes involved in malignant phenotype

To understand the mechanism underlying the function of HOXC6, we investigated the downstream targets identified by RNA-seq. As demonstrated by GO analysis (Figs. 7H and 7I), the genes regulated by HOXC6 could be categorized into various functional groups, including organelle fission and nuclear division in Eca109-HOXC6 cells, and angiogenesis and tRNA aminoacylation for protein translation in TE10-HOXC6 cells. Functional pathway analysis (Figs. 7J and 7K) suggested that HOXC6 may have functions mediated by crosstalk with important signaling pathways such as p53 and focal adhesion in Eca109-HOXC6 cells, and the TGF-β signaling pathway and aminoacyl-tRNA biosynthesis in TE10-HOXC6 cells. These analyses indicate that HOXC6 executed its function via distinct mechanisms in various cell lines. However, these results failed to provide clear clues to how HOXC6 affects the malignant phenotype of cells. We then investigated the genes up-regulated in both Eca109-HOXC6 and TE10-HOXC6 cells. Interestingly, as indicated in Table 3, there were many genes involved in the malignant phenotype of various cancers. Furthermore, as demonstrated by bioinformatic analysis (Fig. 8), some of the genes up-regulated by HOXC6 were highly expressed in ESCC. In the present study, HOXC6 further up-regulated the expression of these genes. This evidence suggests that HOXC6 may execute its function via activating the expression of genes involved in malignant phenotype.

Table 3 List of genes regulated by HOXC6 in Eca109 and TE10 stable cells.

Gene	Description of function	Cancer type	Reference	
Eca109-up	MMP14	Promote cell migration and invasion	Nasopharyngeal carcinoma	Yan et al. (2015)	
Promote tumor invasion and angiogenesis	Pituitary adenomas	Hui et al. (2015)	
SATB1	Promote cell growth and invasion	pancreatic cancer	Chen et al. (2015)	
Promote cell growth and invasion	Prostate cancer	Mao et al. (2013)	
Promote cell invasion and metastasis	Breast cancer	Han et al. (2008)	
AKAP9	Promote proliferation, migration and invasion	Colorectal cancer	Yang et al. (2015)	
Promote proliferation, migration and invasion	Colorectal cancer	Hu et al. (2016)	
CFH	Promote proliferation and migration	Cutaneous squamous cell carcinoma	Riihila et al. (2014)	
Promote tumorsphere formation	Liver cancer	Seol et al. (2016)	
JAK2	JAK2 inhibition prevents cell migration and invasion	Glioblastoma	Senft et al. (2011)	
JAK2 inhibition suppresses cell migration, invasion and proliferation	Cervical cancer	Luo et al. (2016)	
TE10-up	SPARC	Promote cell invasion and growth	Gastric cancer	Yin et al. (2010)	
	Promote cell proliferation, invasion and metastasis, induced cell apoptosis	Ovarian cancer	Chen et al. (2012a)	
FN1	Promote proliferation, migration and invasion	Thyroid cancer	Sponziello et al. (2016)	
Promote proliferation, migration and invasion	Gastric cancer	Zhang et al. (2017)	
WNT6	Promote proliferation, cell cycle and migration, but inhibit cell apoptosis	Colon cancer	Zheng & Yu (2018)	
Inhibit cell apoptosis	Gastric cancer	Yuan et al. (2013)	
Common-up	SEMA3C	Promote tumor growth and metastasis	Pancreatic ductal adenocarcinoma	Xu et al. (2017)	
Promote the survival and tumorigenicity of glioma stem cells	Glioma	Man et al. (2014)	
Promote adhesion, invasion and proliferation	Breast cancer	Malik et al. (2016)	
	INHBA	Promote cell proliferation	Lung adenocarcinoma	Seder et al. (2009b)	
Promote cell proliferation	Esophageal adenocarcinoma	Seder et al. (2009a)	
SGK1	Promote cell proliferation and migration	Colorectal cancer	Liang et al. (2017)	
Promote cell proliferation	Colorectal cancer	Lang, Perrotti & Stournaras (2010)	
HEY1	Promote cell invasion and metastasis	Osteosarcoma	Tsuru et al. (2015)	
Promote proliferation	Glioblastoma	Hulleman et al. (2009)	

Figure 8 Gene expression omnibus database analysis.

Many genes regulated by HOXC6 in ESCC such as (A–B) MMP14, (C–D) SPARC, (E–F) FN1, (G–H) INHBA, (I–J) SEMA3C and (K–L) HEY1 were highly expressed in ESCC.

Discussion

Tumor development is often associated with the abnormal expression of critical genes (Hanahan & Weinberg, 2011; Wang et al., 2013). HOXC6 belongs to the HOX gene family and encodes HD-containing transcription factors with the capacity to bind specific DNA sequences and regulate the expression of downstream genes (Hussain et al., 2015). The aberrant expression of HOXC6 has been reported in many cancer types; however, the mechanisms underlying the function of this gene in cancer cells have not been fully elucidated. We explored the role of HOXC6 in the malignant phenotype of ESCC. Based on our results, the expression of HOXC6 was significantly elevated in ESCC cells. In addition, ectopic expression of HOXC6 promoted the migration, invasion, and proliferation of ESCC cells. Since these phenotypes are directly related to the development and progression of cancer, HOXC6 emerges as an oncogene in ESCC. This is consistent with the results of other reports focused on the function of HOXC6 in cancers other than ESCC.

Like other HD proteins, HOXC6 may execute its effects via binding to specific sequences in the genome following HOXC6 transfection. However, to our surprise, we found that HOXC6 exerted its effects on the transcription patterns of cancer in a cell-context-dependent manner. HOXC6 modulated distinct sets of genes in different ESCC cell lines. This may result from the fact that the specificity of an HD protein usually requires the formation of various complexes, and the availability of other cofactors may be critical for the modulation of downstream targets (Ladam & Sagerstrom, 2014). Many factors have been identified as interacting partners of homeobox-containing gene products. For example, these factors may interact with other members of the HD protein family, chromatin remodeling factors, or other transcription factors (Ladam & Sagerstrom, 2014). The function of an HD protein may be influenced by both the expression level and the modification status of these interacting partners.

To determine the mechanisms underlying the oncogenic function of HOXC6, we compared the genes that were up-regulated in Eca109-HOXC6 and TE10-HOXC6 cells. Results showed that both of these cell lines contain genes with unambiguous functions that are associated with the malignant phenotype in various types of cancers. These genes included MMP14, SPARC, and FN1. MMP14 is a member of the matrix metalloproteinase (MMP) family, which can degrade collagen and other extracellular matrix proteins (Ulasov et al., 2014). MMP14 not only promotes cell migration, invasion, and angiogenesis in nasopharyngeal carcinoma (Yan et al., 2015) and pituitary adenomas (Hui et al., 2015) but also promotes the secretion of pro-MMP2 and pro-MMP9 (Zarrabi et al., 2011). SPARC is a collagen-binding glycoprotein that interacts with MMPs and growth factors, such as TGF-β and fibroblast growth factor (Vaz et al., 2015). SPARC can enhance cell invasion, metastasis, and growth while inducing apoptosis in gastric (Yin et al., 2010) and ovarian (Chen et al., 2012a) cancers. FN1 can induce abnormal expression of some MMPs, such as MMP9/MMP2 (Moroz et al., 2013; Qian et al., 2011) and promote proliferation, migration, and invasion in thyroid (Sponziello et al., 2016) and gastric (Zhang et al., 2017) cancers. In addition, AKAP9, SATB1, SEMA3C, SGK1, and INHBA, etc. have also been reported to enhance cell migration, invasion, proliferation and angiogenesis or induce apoptosis in various cancers, such as pancreatic (Chen et al., 2015; Xu et al., 2017), glioma (Hulleman et al., 2009; Man et al., 2014; Senft et al., 2011), breast (Han et al., 2008; Malik et al., 2016), lung (Seder et al., 2009b), esophageal (Seder et al., 2009a), prostate (Mao et al., 2013), and colorectal (Hu et al., 2016; Lang, Perrotti & Stournaras, 2010; Liang et al., 2017; Yang et al., 2015; Zheng & Yu, 2018), cutaneous squamous cell carcinoma (Riihila et al., 2014), liver (Seol et al., 2016), cervical (Luo et al., 2016), gastric (Yuan et al., 2013) and osteosarcoma (Tsuru et al., 2015) cancers.

In conclusion, HOXC6 promoted ESCC cell migration, invasion, and proliferation, and its function may be related to the aberrant expression of genes caused by HOXC6 overexpression. HOXC6 may be a new significant biomarker for diagnosis, therapy, and prognosis. Targeted inhibition of HOXC6 may be a new strategy for the treatment of ESCC. However, the precise molecular mechanism is not completely understood, and further investigation is still needed.

Conclusion

HOXC6 promoted the malignant phenotypes of ESCC cells. HOXC6 could activate the expression of oncogenic genes in a cell context-dependent manner. Targeted inhibition of HOXC6 might provide a new strategy for the therapy of ESCC.

Supplemental Information

Supplemental Information 1 The raw numeric data and images from growth curves, wound-healing/cell invasion/cell proliferation assays and gene expression experiments.

Click here for additional data file.

We offer many thanks to all colleagues for their contribution to this study.

Additional Information and Declarations

Competing Interests

Author Contributions

Human Ethics

Data Availability

The authors declare that they have no competing interests.

Li Tang performed the experiments, analyzed the data, contributed reagents/materials/analysis tools, prepared figures and/or tables, authored or reviewed drafts of the paper, approved the final draft.

Yong Cao performed the experiments, approved the final draft.

Xueqin Song performed the experiments, approved the final draft.

Xiaoyan Wang performed the experiments, contributed reagents/materials/analysis tools, approved the final draft.

Yan Li performed the experiments, approved the final draft.

Minglan Yu performed the experiments, approved the final draft.

Mingying Li performed the experiments, approved the final draft.

Xu Liu performed the experiments, analyzed the data, contributed reagents/materials/analysis tools, prepared figures and/or tables, approved the final draft.

Fang Huang performed the experiments, approved the final draft.

Feng Chen conceived and designed the experiments, approved the final draft.

Haisu Wan conceived and designed the experiments, authored or reviewed drafts of the paper, approved the final draft.

The following information was supplied relating to ethical approvals (i.e., approving body and any reference numbers):

The Ethics Committee of the Affiliated Hospital of Southwest Medical University provided approval (k2018002-r) to carry out the study within its facilities.

The following information was supplied regarding data availability:

GEO accession GSE121976.

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
