# Peer review of "HOXC6 promotes migration, invasion and proliferation of esophageal squamous cell carcinoma cells via modulating expression of genes involved in malignant phenotypes"

_PeerJ, doi:10.7717/peerj.6607_

## Round 0.1 · original submission · Major Revisions

Although it is of interest, we are unable to consider it for publication in its current form. The reviewers have raised a number of points which we believe would improve the manuscript and may allow a revised version to be published in PeerJ. If you are able to fully address these points, we would encourage you to submit a revised manuscript to PeerJ. Once you have made the necessary corrections, please submit online. I look forward to receiving your revised manuscript soon.

Reviewer 1 ·

Basic reporting

no comment

Experimental design

no comment

Validity of the findings

no comment

Additional comments

To investigating the role of HOXC6 in esophageal squamous cell carcinoma (ESCC), authors examined the expression of HOXC6 in ESCC tissues by immunohistochemistry, quantitative real-time PCR and immunoblotting assays, and found that HOXC6 was highly expressed in ESCC tissues. Furthermore, HOXC6 overexpression promoted the migration, invasion, and proliferation of both Eca109 and TE10 cells. Further bioinformatics analysis showed that HOXC6 activated several crucial genes implicated in the malignant phenotype of cancer cells. These results were meaningful and would contribute to understanding of ESCC pathogenesis. However, there is a question as follows.
1. In order to enhance reproducibility and allow for new discoveries, raw RNA-seq data should be uploaded to public database, such as Sequence Read Archive (SRA). Accession number should be supplied in the manuscript.

Reviewer 2 ·

Basic reporting

In this manuscript, Tang Li et al., aim to demonstrate a function for the HOXC6 protein in esophageal squamous cell carcinoma (ESCC). Overall, the manuscript is well structured, and provides sufficient evidence to suggest a potential proto-oncogenic for HOXC6 in ESCC. A few minor revisions could help strengthen the quality of the manuscript:

-In figure 1, the authors should provide more detail on the precise tumor data set used. Specifically, is the “expression” of HOXC6 only considering mRNA levels, or also taking into account copy number alterations? Additionally, none of the y axis on the blots are labelled, making it harder to interpret the data

-In figure 2C, do each of the columns in the bar graph represent a single patient derived tumor sample? If so, this should be specified in the legend.

Experimental design

- One of the more intriguing experimental issues with this study is the choice of studying the effect of HOXC6 overexpression in ESCC cell lines. Given that ESCC is associated with higher expression of HOXC6 than normal to begin with (as demonstrated in figure 2), presumably studying the effect of knocking down HOXC6 would be more meaningful. While the overexpression data seems convincing, it would be helpful to know if the authors performed knockdown experiments in the same cell lines, even if the results were not conclusive

-In the wound healing assays (figure 4), it is unclear what metric is exactly being measured in the graphs in panels C and D. Specifically, is this the width of the wound gap, or the recovery width? These values are inversely correlated. The images would indicate it is the latter, but this needs to be clearly specified by the authors

Validity of the findings

Overall, I believe the data presented in this manuscript support the conclusions made by the authors. However, the authors should avoid using broad language that may over interpret the data. For example, in line 194, the authors state that “HOXC6 expression is positively correlated with the progression of ESCC”. Drawing such a conclusion would require a time-course study of the expression of HOXC6 during progression of ESCC. A single snapshot of expression data is not sufficient to make conclusions regarding progression of the tumor.

Reviewer 3 ·

Basic reporting

The manuscript #32037 by Tang et. al. use established cell lines and tissue samples and experimental methods including RNA-seq to show that HOXC6 promotes migration, invasion and proliferation of esophageal squamous cell carcinoma (ESCC). The study is not novel, however, the authors repeat the results in this carcinoma cells and establish it.

Figure legend should be made more descriptive. Number of experiment (N) should be mentioned for each graph in the figure legend.

Human participant consent form is in a language other than English, probably in Chinese.

Experimental design

The authors use ESCC lines - Eca-109 and TE-10 throughout the study however comparison to normal (non-carcinoma cells) is missing. I suggest that the authors compare their findings in Eca-109 and TE-10 with control cell line -Het-1A (ATCC).

Authors validate the role of HOXC6 by overexpressing it in Eca-109 and TE-10. However, since these are carcinoma cell line and have already alleviated levels of HOXC6. Hence a comparison with Het-1A (ATCC) is a must and also knock down of HOXC6 will further support the results.

The authors mention about 20 similarly regulated genes (17- upregulated and 3 down regulated) upon HOXC6 overexpression in the two cell line. These should be provided as a table. Table 3 mentions some of these but the full data should be provided.

Authors should check the cell to cell HOXC6 overexpression variability in the two cell lines. The difference could arise due to different expression level of HOXC6. HOXC6 should be immunostained and intensity per cell from multiple cells should be compared between ECa-109 and TE-10 overexpression cell lines.

Validity of the findings

In fig 1, the source and paper for each data should be explicitly mentioned either in methods section or figure legend. Details of the bioinformatics analysis should also be provided.

In fig2C, how was the histogram obtained. The authors should mention it in the figure legend.

In fig2D, the band size should be mentioned next to bands and quantification of western should be carried out and plotted in a figure.

In fig3. qRT-PCR and western should be carried out for control proteins like vinculin or any housekeeping gene.

In, Fig. 4C, 4D, 5C, 5D, 6A (both graphs), significance should be tested for all time points not just the last time point. Wherever the two curves become significantly different, the time point should be mentioned in the graph or in the figure legend.

In fig 6B (right graph), it appears that OD450 value for TE10-NEO (red curve) was carried out just once. There is no error bar.

Additional comments

Is there any repository for RNA -seq data? Link should be mentioned in the paper.

---

## Round 0.2 · accepted · Accept

The authors have worked hard to improve the quality of their manuscript. And the quality of the revised manuscript is quite good. My suggestion is that it can be published.

# Reviewer 2 ·

Basic reporting

The author's have addressed my comments, and I believe the article is suitable for publication. I have no further comments

Experimental design

The author's have addressed my comments, and I believe the article is suitable for publication. I have no further comments

Validity of the findings

The author's have addressed my comments, and I believe the article is suitable for publication. I have no further comments